# The Perception of the Patient Safety Climate by Health Professionals during the COVID-19 Pandemic—International Research

**DOI:** 10.3390/ijerph19159712

**Published:** 2022-08-06

**Authors:** Justyna Kosydar-Bochenek, Sabina Krupa, Dorota Religa, Adriano Friganović, Ber Oomen, Elena Brioni, Stelios Iordanou, Marcin Suchoparski, Małgorzata Knap, Wioletta Mędrzycka-Dąbrowska

**Affiliations:** 1Institute of Health Sciences, Medical College, Rzeszow University, Warzywna St. 1, 35-310 Rzeszow, Poland; 2Division for Clinical Geriatrics, Department of Neurobiology, Care Sciences and Society (NVS), Karolinska Institutet, 14152 Huddinge, Sweden; 3Department of Anesthesiology and Intensive Medicine, University Hospital Centre Zagreb, 10000 Zagreb, Croatia; 4Department of Nursing, University of Applied Health Sciences, Mlinarska Cesta 38, 10000 Zagreb, Croatia; 5The European Specialist Nurses Organisation (ESNO), 6821HR Arnhem, The Netherlands; 6Nephrology and Dialysis Unit, San Raffaele Hospital, 20132 Milan, Italy; 7Intensive Care Unit, Limassol General Hospital, Kato Polemidia 3085, Cyprus; 8Admission Room District Hospital in Golub Dobrzyń, 87-400 Golub Dobrzyń, Poland; 9Institute of Health Sciences, Collegium Medicum, Jan Kochanowski University of Kielce, 25-369 Kielce, Poland; 10Department of Anaesthesiology Nursing & Intensive Care, Faculty of Health Sciences, Medical University of Gdansk, 80-211 Gdansk, Poland

**Keywords:** safety climate, safety culture, safety attitudes questionnaire, healthcare workers, health professionals, nurse, physician, paramedic

## Abstract

The patient safety climate is a key element of quality in healthcare. It should be a priority in the healthcare systems of all countries in the world. The goal of patient safety programs is to prevent errors and reduce the potential harm to patients when using healthcare services. A safety climate is also necessary to ensure a safe working environment for healthcare professionals. The attitudes of healthcare workers toward patient safety in various aspects of work, organization and functioning of the ward are important elements of the organization’s safety culture. The aim of this study was to determine the perception of the patient safety climate by healthcare workers during the COVID-19 pandemic. Methods: The study was conducted in five European countries. The Safety Attitude Questionnaire (SAQ) short version was used for the study. A total of 1061 healthcare workers: physicians, nurses and paramedics, participated in this study. Results: All groups received the highest mean results on the stress recognition subscale (SR): nurses 98.77, paramedics 96.39 and physician 98.28. Nurses and physicians evaluated work conditions (WC) to be the lowest (47.19 and 44.99), while paramedics evaluated perceptions of management (PM) as the worst (46.44). Paramedics achieved statistically significantly lower scores compared to nurses and physicians in job satisfaction (JS), stress recognition (SR) and perception of management (PM) (*p* < 0.0001). Paramedics compared to nurses and physicians rank better in working conditions (WC) in relation to patient safety (16.21%). Most often, persons of lower seniority scored higher in all subscales (*p* = 0.001). In Poland, Spain, France, Turkey, and Greece, healthcare workers scored highest in stress recognition (SR). In Poland, Spain, France, and Turkey, they assessed working conditions (WC) as the worst, while in Greece, the perception of management (PM) had the lowest result. Conclusion: Participant perceptions about the patient safety climate were not at a particularly satisfactory level, and there is still a need for the development of patient safety culture in healthcare in Europe. Overall, positive working conditions, good management and effective teamwork can contribute to improving employees’ attitudes toward patient safety. This study was carried out during the COVID-19 pandemic and should be repeated after its completion, and comparative studies will allow for a more precise determination of the safety climate in the assessment of employees.

## 1. Introduction

Patient safety is an important aspect of healthcare delivery and quality of service; therefore, it should be a priority in any healthcare system in the world. There are numerous definitions of patient safety in the literature. According to the World Health Organization (WHO): “patient safety means the reduction to an acceptable minimum level of risk of unnecessary harm related to health care” [1]. The agency Safety Improvement for Patients in Europe asserted that patient safety focuses on identifying, analyzing, and minimizing patient risk [2,3]. The safety climate, defined as the collective perception by workers of the importance of safety in their organization, is related to the safety and performance of patients and healthcare workers, as well as to injuries, exposure, and safe work compliance among medical personnel [4]. In healthcare, the safety climate is particularly important due to the specific nature of healthcare facilities and the related responsibilities of employees to protect patients from harm while using services. Patient safety is viewed as a crucial component of quality in healthcare service.

To improve safety climate in healthcare, the WHO established, in 2004, the World Alliance for Patient Safety, aiming at mobilizing global efforts to improve the safety of healthcare for patients in all WHO member countries, setting Patient Safety (PS) in the agenda [5]. The main goal is the development of patient safety culture in health organizations, which can be stimulated in such a way that their workforce and processes can focus on improving the reliability and safety of patient care. The attitudes of health professionals toward patient safety, including the awareness of the risk of adverse events, are important elements of the organization’s safety culture [5]. Therefore, using tools such as the Safety Attitudes Questionnaire (SAQ) for assessing healthcare staff attitudes about patient safety at the organizational level is an important part of fostering organizational change to improve patient safety [1,5]. High awareness of occupational safety, teamwork, assessment of the culture of occupational safety, as well as analysis and drawing conclusions, can improve the quality and safety of services and make the patient feel safer [6,7,8]. Research shows that communication and teamwork have a big impact on patient safety [2,3,7,8]. The significant impact on the patient safety climate has reporting errors and safety awareness, gender and demographics, work experience, and staffing levels as well [6].

A safety climate is not only important for reducing the risk for harm to patients but is also key to ensuring a safe working environment for healthcare professionals. A first step toward building safety cultures is to conduct research to assess healthcare workers’ attitudes toward patient safety [1]. The current knowledge about the patient safety culture in the healthcare system and the research tools used to assess the safety climate in hospitals is insufficient, which may hinder current efforts to improve patient safety worldwide [6].

## 2. Materials and Methods

### 2.1. Study Design

A cross-sectional study was conducted. The aim of the study was to evaluate the patient safety attitudes of healthcare workers during the COVID-19 pandemic in selected European countries.

The following detailed objectives were defined in the study:

What attitudes toward factors related to hospitalized patient safety are presented by nurses, physicians and paramedics?What were the differences in attitudes toward safety in the study groups?To what extent did the place of residence differentiate the attitudes toward safety of nurses, physicians, and paramedics?What was the relationship between the time of employment of the respondents and their attitudes toward safety?

### 2.2. Characteristics of the Research Tool

The study used the short version of the Safety Attitude Questionnaire (SAQ-SF) to evaluate attitudes of nurses, paramedics, and physicians toward factors of patient safety.

The SAQ was developed by a team of researchers from the University of Texas [9]. The SAQ contains six dimensions: teamwork climate, safety climate, job satisfaction, stress recognition, perception of management, and working conditions. It has been adapted for use in different settings; these alternate versions include the ICU version, Ambulatory version, Operating Room version, Pharmacy version, Labor and Delivery version, and Short-Form version. The SAQ uses a five-point Likert scale: 1 = strongly disagree, 2 = slightly disagree, 3 = neutral, 4 = slightly agree, 5 = strongly agree [10].

The present research adopted the generic SAQ Short-Form version (available at https://med.uth.edu/chqs/survey/ accessed on 5 May 2022). This version consists of 36 items, 31 of which are categorized into 6 dimensions and 5 of which do not belong to any dimension. The items are as follows: teamwork climate: items 1–6; safety climate: items 7–13; job satisfaction: items 15–19; stress recognition: items 20–23; perceptions of management: items 24–29 (each of these items is measured at two levels: unit and hospital); and working conditions: items 30–32. items 14 and 33–36 are not part of the subscales above; items 2, 11, 20–23 and 36 are reverse-scored. To calculate the percentage of respondents who are positive (i.e., percent agreement), evaluate the percentage of respondents who obtained a scale score of 75 or higher. A score of 75 on the scale score indicates the same thing as “agree slightly”, which corresponds to the number 4 on the original 5-point Likert scale [9].

Teamwork climate (TS) is understood as perceived quality of collaboration between personnel. Safety climate (SC) is the perceptions of a strong and proactive organizational commitment to safety. However, job satisfaction (JS) is positivity about the work experience. Stress recognition (SR) is acknowledgement of how performance is influenced by stressors. Perceptions of management (PM) is basically the approval of managerial action. Working conditions (WC) perceive quality of the work environment and logistical support (staffing, equipment, etc.) [11].

The Safety Attitudes Questionnaire demonstrated good psychometric properties. Composite scale reliability for the SAQ was assessed via Raykov’s ρ coefficient. The ρ value for the SAQ in this sample was 0.90, indicating strong reliability of the SAQ [11].

### 2.3. Data Collection, Setting and Procedure

The study was conducted from 10 March to 10 May 2022 as an online survey. Before starting the study, the authors of the article obtained consent from the authors to use the tools. The survey was addressed to medical personnel: nurses, paramedics, and physicians. The questionnaire was posted on the websites of international associations of medical personnel and sent to the e-mail addresses of the members of these associations. We chose an online questionnaire for the survey because face-to-face testing was impractical given the COVID-19 pandemic and the risk of contamination. Furthermore, the online survey was quick, easy, and convenient to collect and analyze the data. The survey was in English. A short presentation informed participants about the objectives of the study, followed by a survey. At the end, participants completed a short demographic questionnaire. To guarantee anonymity, no personal data were required that would enable the identification of the respondents. By completing the questionnaire, the participants expressed their consent to participate in the study, but they could withdraw from the study at any stage of completing the form. It took about 10 min to complete the entire questionnaire. Participants were informed that participation was voluntary and anonymous, that all responses would be kept confidential and that no individual responses would be available to hospital management.

### 2.4. Inclusion and Exclusion Criteria

**Inclusion criteria:** healthcare workers; adults; people who are working as a physician, nurse, or paramedic; knowledge of the English language; work in the hospital for at least 4 weeks, professionally active.

### 2.5. Ethical Considerations

This study was approved by the Bioethics Committee of the University of Rzeszow (KBE No. 2022/013). The authors followed the guidelines of the Declaration of Helsinki (World Medical association, 2013). To guarantee anonymity, no personal data permitting the identification of the respondents were required. The participants could withdraw from the survey at any moment without providing any justification, and no data were saved. 

The study was conducted according to the Strengthening the Reporting of Observational Studies in Epidemiology (STROBE) criteria.

### 2.6. Statistics

Descriptive analysis was used to characterize the study sample in terms of demographic information. The Pearson (r) correlation was calculated between the subscales of the SAQ questionnaire. For questions in individual SAQ scales, % of neutral, disagree and agree cases were calculated, as well as descriptive mean statistics and standard deviations. For six SAQ scales, the mean and standard deviations as well as kurtosis and skewness were calculated on the basis of the kurtosis and skewness values from the range (−1, 5, 1, 5); the distributions of these rocks were adopted as normal distributions. Pearson’s correlation (r) was calculated between the subscales of the SAQ questionnaire. The level of significance was *p* < 0.05. In order to check the influence of gender on the value of SAQ subscales, the Student’s t test for two groups was used. The level of significance was *p* < 0.05. To check the influence of position, years in specialty, and country on the value of SAQ subscales, the ANOVA (F) test was used. To check the significance of intergroup differences, a post hoc test, the NIR test, was used. The level of significance was *p* < 0.05. The significance of the distribution of positive and negative results for the SAQ subscales in gender, position, years in specialty, and country groups was checked using the chi-square test (χ^2^). The level of significance was *p* < 0.05. Multiple regression analysis (progressive stepwise regression) was performed. Analyses were performed using TIBCO Software Inc. (2017). Statistica (data analysis software system), version 13 (StatSoft, Tulsa, OK, USA; TIBCO Software, Palo Alto, CA, USA; Dell, Austin, TX, USA) [12]. 

## 3. Results

### 3.1. Characteristics of the People Participating in the Study

The questionnaire was completed by 1454 people. In total, 1061 questionnaires were included in the analysis (393 questionnaires were not complete). In total, 905 women (78%) and 255 (22%) men took part in the study. The average age was 38 years (min 23-max 60). The analyses included surveys of residents of the five countries that took part in the study the most. Respondents came from five European countries: France (21.7%), Greece (10.9%), Poland (11.8%), Spain (31.3%) and Turkey (24.3%). Respondents were mostly nurses at 561 (52.9%), 253 paramedics (23.8%) and 247 physicians (23.3%). Healthcare workers with 3–4 year of experience were the largest group, whereas the smallest groups had experience in the range of 6 to 11 months or 11 years and more. Details on participating healthcare worker characteristics are depicted in Table 1.

### 3.2. Results of Safety Attitude Questionnaire (SAQ-SF) in Research Group

For teamwork climate positive responses ranged from 0.0% to 91.4%; for safety climate from 0.0% to 94.3%; for job satisfaction from 0.0% to 84.0%; for stress recognition from 78.0% to 100.0%; for perceptions of unit management from 14.1% to 100.0%; and for working conditions from 21.8% to 70.8% (Table 2).

All groups received the highest mean results on the stress recognition subscale (SR): nurses 98.77, paramedics 96.39 and physician 98.28. Nurses and physicians evaluated work conditions (WC) as the lowest (47.19 and 44.99), while paramedics evaluated perceptions of management (PM) as the worst (46.44), as shown in Table 3.

The analysis showed statistically significant differences between the groups of nurses, paramedics, and physicians in all six subscales, delineating different aspects of the evaluation of attitudes toward factors fostering patient safety. The nurses obtained higher mean results in teamwork climate (TC) compared with paramedics and physicians (51.57 vs. 50.69 and 50.51; *p* < 0.001), while higher mean results in safety climate (SC) were obtained paramedics and physicians compared to nurses (83.26 and 81.98 vs. 79.40; *p* < 0.0001). Lower mean results in job satisfaction (JS) were obtained in group of paramedics compared to nurses and physician (84.90 vs. 87.67 and 88.81; *p* < 0.0001). In subscale stress recognition (SR), the lowest mean statistically significant result was also obtained among paramedics compared to nurses and physicians (96.39 vs. 98.77 and 98.28; *p* < 0.0001). This was similar in the case of perceptions of management (PM): (46.44 vs. 53.31 and 50.73; *p* < 0.0001). There was also a statistically significant difference between the group of doctors and nurses: mean results were lower in the group of physicians rather than in nurses (*p* < 0.001). The paramedics obtained higher mean results in working condition (WC) compared with nurses and physicians (54.15 vs. 47.19 and 44.99; *p* < 0.001). The results are shown in Table 3.

For teamwork climate (TC), the mean result in the group of women was significantly higher than the mean values in the group of men. For job satisfaction (JS), stress recognition (SR) and perception of management (PM), the mean values for women were significantly lower than the mean values of these scales for men. No statistically significant differences were found for the mean values of women and men in the safety climate (SC) and working conditions (WC). The results are shown in Table 3.

A statistically significant difference was shown between the professional experience of the surveyed staff and five subscales, except for teamwork climate (TC). Most often, persons of lower seniority scored higher in all subscales. Mean score in nearly all subscales was significantly higher for those who work less than 6 months and between 6 and 11 months. Only in the case of perception of management (PM) and working conditions (WC) was the mean result, in the group of healthcare workers with seniority of 6–11 months, significantly lower than in the group of people working longer. The results are shown in Table 3.

In Poland, healthcare workers scored highest in stress recognition (SR) (98.40), while they evaluated working conditions (WC) as the lowest (48.90). In France, similarly, healthcare workers scored highest in stress recognition (SR) (97.83), while they evaluated working conditions (WC) as the lowest (48.10). In Greece, healthcare workers scored highest in stress recognition (SR) (100.00), while they evaluated perception of management (PM) as the lowest (47.33). In Spain, healthcare workers scored highest in stress recognition (SR) (97.82), while they evaluated working conditions (WC) as the lowest (47.29). In Turkey, healthcare workers scored highest in stress recognition (SR) (97.67), while they evaluated working conditions (WC) as the lowest (47.09). The results are shown in Table 3.

A statistical relationship was observed between country and teamwork climate (TC), stress recognition (SR), perception of management (PM) and working conditions (WC). Mean score in the teamwork climate (TC) subscale was significantly lower in France (49.09 ± 4.17) than in Greece (51.36 ± 3.19), Poland (55.17 ± 5.81), Spain (50.11 ± 4.18) and in Turkey (52.13 ± 3.29). The mean result of teamwork climate (TC) subscale was significantly lower in Greece (51.36 ± 3.19) than in Poland (55.17 ± 5.81). At the same time, it was significantly higher in Poland (55.17 ± 5.81), than in Spain (50.11 ± 4.18) and in Turkey (52.13 ± 3.29). The mean result of the teamwork climate (TC) subscale was significantly lower in Spain (50.11 ± 4.18) than in Turkey (52.13 ± 3.29). Mean score in the stress recognition (SR) subscale was significantly higher in Greece (100.0 ± 0.00) than in France (97.83 ± 4.75), Spain (97.82 ± 4.75), Turkey (97.67 ± 4.87) *p* < 0.001 and in Poland (98.40 ± 4.19 *p* < 0.01). The mean score in the perception of management (PM) subscale was significantly lower in Greece (47.33 ± 9.77) than in France (50.98 ± 11.57), Poland (51.88 ± 11.35) and Spain (53.42 ± 9.01). In turn, the average result in working conditions (WC) was significantly higher in Greece (49.09 ± 4.17), than in France (48.10 ± 12.52), Spain (47.29 ± 14.22) and in Turkey (47.09 ± 17.26). The results are shown in Table 3 and Figure 1.

The distribution of positive (score ≥ 75 points) and negative (score < 75 points) results depending on the type of professional group for the subscales safety climate (SC), job satisfaction (JS), perceptions of management (PM) and working conditions (WC) was found to be significant.

The largest percentage of nurses, paramedics and physicians showed positive attitudes (score ≥ 75 points) in relation to the safety of patients, toward stress recognition (SR): 100% vs. 100% vs. 97.63%. A large percentage of nurses, paramedics and physicians showed positive attitudes (score ≥ 75 points) in relation to job satisfaction (JS): 89.3% vs. 90.51% vs. 93.68%. Perception of management (PM) received the lowest scores in all study groups (0.0% to 2.14%). Working conditions (WC) received low scores (from 2.37% to 16.21%). Working conditions in relation to patient safety paramedics rated the best (16.21%). The results are shown in Table 4.

The distribution of positive (score ≥ 75 points) and negative (score < 75 points) results in groups were divided according to country for the subscales safety climate (SC), job satisfaction (JS), perceptions of management (PM) and working conditions (WC).

The greatest differences were observed for the working conditions (WC) subscale. Positive attitudes (score ≥ 75 points) in relation to the safety of patients toward working condition was low, with the lowest in France (0%) and the highest in Greece (26.72%). The results are shown in Table 5.

### 3.3. Correlation and Multiple Regression Analysis

Pearson’s (r) correlation was calculated between the SAQ subscales. The results are presented in Table 6.

At a later stage, how the SAQ subscale model is influenced by subsequent variables was checked: country, position, and years. The independent variables were dichotomous variables of individual countries, occupation, and seniority, while the dependent variables were the variables of the SAQ subscales. Then, multiple regression analysis (progressive stepwise regression) was performed. As a result of the analysis, well-fitted models at the level of *p* < 0.001, explaining 7% to 19% of the results, were obtained. The results are presented in Table 7.

## 4. Discussion

This study examines healthcare professionals’ attitudes toward patient safety based on a short version of the SAQ-F. In the stress recognition (SR), safety climate (SC) and job satisfaction (JS) SAQ subscales, a positive result, i.e., 75 pts. or more, was obtained by more than 75% of surveyed nurses, paramedics, and physicians. In the perception of management (PM), working conditions (WC) and teamwork climate (TC) SAQ subscales, a positive result, i.e., 75 pts. or more, was obtained by less than one fifth of surveyed nurses, paramedics, and physicians. This finding suggests that the attitudes of healthcare workers of patient safety were not at a particularly satisfactory level, and there is still a need for the development of patient safety culture in healthcare in Europe.

Our study showed that Polish healthcare workers scored highest in stress recognition (SR) (98.40), while they evaluated working conditions (WC) as the lowest (48.90). Malinowska-Lipień obtained similar results. In total, 606 nurses and 527 physicians employed in the surgical and medical wards in 21 Polish hospitals around the country participated in the study. Both nurses and physicians scored highest in stress recognition (SR) (71.6 and 80.86), while they evaluated working conditions (WC) as the lowest (45.82 and 52.09). Nurses achieved statistically significantly lower scores compared to physicians in every aspect of the safety attitudes evaluation (*p* < 0.05). Overall, positive working conditions and effective teamwork can contribute to improving employees’ attitudes toward patient safety [1].

As the results of our study show, healthcare workers in Turkey scored highest in stress recognition (SR) (97.67), while they evaluated working conditions (WC) as the lowest (47.09). Previous research in Turkey showed that the most important factor affecting patient safety attitudes was teamwork climate [13,14]. Other study showed that the attitudes among cardiology and cardiovascular surgery nurses working in a Turkish facility toward patient safety were not at a particularly satisfactory level. Moreover, the cardiology nurses were found to have a more positive attitude toward patient safety than their colleagues in cardiovascular surgery [15]. The study that was aimed at evaluating the patient safety attitudes of Turkish surgical nurses showed that none of the six domains of safety culture, including job satisfaction, teamwork, safety climate, the perception of management, stress recognition, and working conditions, achieved a positive mean score over 75 [16]. Nurses who previously received training on patient safety had statistically higher attitude scores than those who did not (U = 3883.000; *p* = 0.01). Nurses working in surgical units had a positive attitude toward patient safety, and previous training on patient safety significantly improved their attitude scores. A recommendation is to conduct effective in-service training programs for patient safety in hospitals and to encourage participation by nurses in training programs such as courses and conferences that will result in attitude improvement [17]

Our study showed that healthcare workers in Greece scored highest in stress recognition (SR) (100.00), while they evaluated perceptions of management (PM) as the lowest (47.33). Attitudes toward the safety climate among medical personnel were also studied among healthcare workers in Greece [18]. Considerable safety climate variations between the ICUs of the regional hospitals of Cyprus have been verified. Age, infrastructure, the severity of cases, and the nurse skill mix are variables that affect the patient safety culture in an ICU environment [19].

The results of our study showed that healthcare workers in France scored highest in stress recognition (SR) (97.83), while they evaluated working conditions (WC) as the lowest (48.10). Similarly, in Spain, healthcare workers scored highest in stress recognition (SR) (97.82), while they evaluated working conditions (WC) as the lowest (47.29). The scores in the teamwork climate (TC) and safety climate (SC) subscales were also similar in France and Spain: (France: TS 49.09 and SC 80.73; Spain: TS 50.11 and SC 81.36). The results of Kristensen’s [20] research from seven European countries, including France and Spain, showed that teamwork climate (TS) was reported as positive by 43% of frontline clinicians. Safety climate (SC) was perceived as positive by 32% of frontline clinicians. Furthermore, positive associations were found between implementation of quality management systems and teamwork and safety climate.

When comparing the findings internationally with countries outside of Europe, the analyses showed that surveyed healthcare workers received higher mean results in the stress recognition subscale than healthcare workers working in China [21], Saudi Arabia [22], the United States of America [23] and Australia [24]. In our study, health workers from Europe obtained lower results in the working conditions (WC) subscale than nurses working in China [21], Saudi Arabia [22] and Australia [24] or in Brazil [25]. Average results in the case of job satisfaction (JS), safety climate (SC) and perception of management (PM) were similar. Surveyed healthcare workers received lower mean results in the teamwork climate (TC) subscale than healthcare workers working in Asian [21], Arabic [22], and in North and South America countries [23,25], as well as Australia [24].

Comparing the results of the research with the research carried out in other European countries—in Sweden, the attitudes of nurses of safety climate in the stress recognition (SR) subscale was lowest (66.1–69.9) [26]; which was similar in Norway (78.0) [27]. The mean score of working conditions (WC) was higher in Norway than in our group (72.7 vs. 48.34) [27].

Seniority or years of work experience also influenced the findings in ways consistent with other studies. According to this research, healthcare workers with the shortest seniority demonstrated a better awareness of the negative impact of stress on patient safety. These results are similar to the results obtained by Malinowska-Lipień et al. [1], which documented that the increase in seniority corresponds to the decrease in stress recognition.

Most studies on attitudes of the patient safety climate among group of healthcare workers in the emergency department (ED) included only physicians and nurses [28,29]. There are only a few research studies among paramedics [30]. The review of Alzahrani et al. [31] revealed that the safety attitudes of ED health staff are generally low, especially on teamwork and management support and among nurses when compared to doctors [31]. Our study showed that paramedics received the lowest mean result in stress recognition (SR), job satisfaction (JS) and perception of management (PM), while nurses and physicians scored higher—a difference that was highly significant. In research of healthcare professionals working in the emergency department of a hospital in Brazil, the participants demonstrated satisfaction with their jobs and dissatisfaction with the actions of management regarding safety issues. Participant perceptions about the patient safety climate were found to be negative [30]. Future research should focus on the safety attitudes of medical staff employed in EDs and pre-hospital rescue and its relationship to medical errors, because paramedics are the “front-line” of hospital care and patients within EDs are especially vulnerable to medical errors.

## 5. Study Limitations

We are aware of several limitations of our study. The SAQ tool relies strongly on self-reported behavior. The resulting information may be biased and may not correctly reflect the actual situation. A positive-answering tendency cannot be excluded. Staff perceptions of communication can vary over time and can be influenced by day-to-day events within the hospital setting. The external validity of the study findings was limited by the study design and participants. Finally, we did not examine variations in perceptions among departments or across hospital settings.

Due to the limitations of the study sample size, these results may not represent the overall data for each country, and the sample should be expanded in future research. Our findings should be tested in future research using a larger independent sample. The relationship between the patient safety climate as perceived by healthcare professionals and patient outcomes should also be explored, as the goal is to build a safe healthcare system and reduce adverse medical events.

## 6. Conclusions

Participant perceptions about the patient safety climate were not at a particularly satisfactory level, and there is still a need for the development of patient safety culture in healthcare in Europe.The results are valuable to identify areas for improvement related to patient safety. Positive working conditions, effective teamwork and better perception of management can contribute to improving employee attitudes toward patient safety.Healthcare leaders should take into account more suggestions from their workers to improve the safety climate in the workplace.This study was carried out during the COVID-19 pandemic and should be repeated after its completion, and comparative studies will allow for a more precise determination of the safety climate in the assessment of employees.

## 7. Implications for Practice

The topic of safety climate at work is not a frequent topic in global research, even though safety is a determinant of both work quality and job satisfaction. It would be right to conduct research not only among European countries, because in-hospital procedures on specific continents can have a huge impact on safety at work. By ensuring proper safety at work, we can count on an increase in willingness to work in each place, an increase in the level of satisfaction, and thus better results of treatment of patients in departments where the safety climate is assessed at a high level. Comparative studies with and after the pandemic can show what has been disturbed in the work of medics and whether it was the pandemic that had a destructive impact on the safety climate, as it can be assumed.

## Figures and Tables

**Figure 1 ijerph-19-09712-f001:**
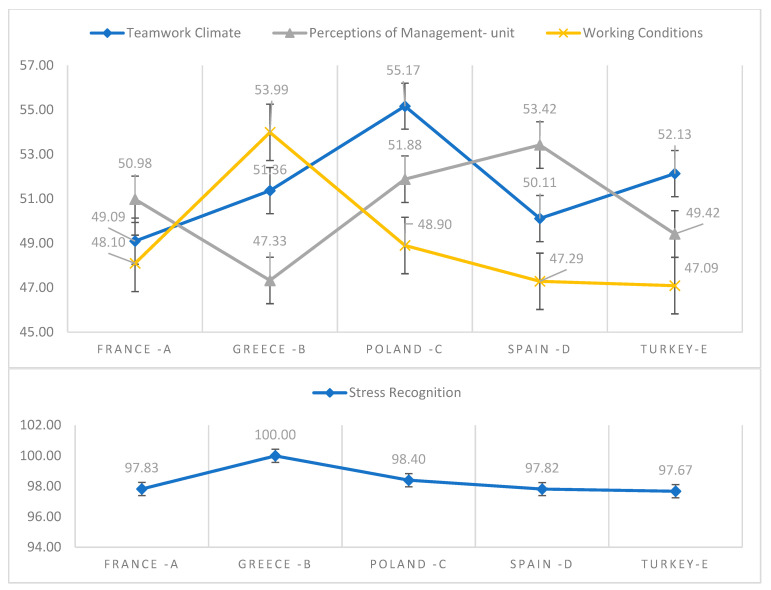
Values of SAQ subscales in various countries.

**Table 1 ijerph-19-09712-t001:** Participant characteristics (*n* = 1061).

Characteristic		*n*	%
Gender	Female Male	828 233	78 22
Age (years)	Under 26 26 to 35 years 36 to 45 years 46 to 55 years 56 years or older	149 357 352 110 93	14 34 33 10 9
Position	Nurse Paramedic Physicians	561 253 247	53 24 23
Years in specialty	Less than 6 months 6–11 months 1 to 2 yrs 3 to 4 yrs 5–10 yrs 11 or more	141 127 208 267 190 128	13 12 20 25 18 12
Country	France Greece Poland Spain Turkey	230 116 125 332 258	20 19 19 20 22

**Table 2 ijerph-19-09712-t002:** The results of the Safety Attitude Questionnaire (SAQ-SF) in the study group.

	% Neutral	% Disagree	% Agree	M	SD
**Teamwork Climate**					
1. Nurse input is well received in this clinical area	0.8%	7.8%	91.4%	4.75	0.82
2. In this clinical area, it is difficult to speak up if I perceive a problem with patient care	0.0%	10.9%	89.1%	4.64	1.05
3. Disagreements in this clinical area are resolved appropriately (i.e., not who is right, but what is best for the patient)	0.0%	100.0%	0.0%	1.88	0.33
4. I have the support I need from other personnel to care for patients	0.0%	0.0%	100.0%	5.00	
5. It is easy for personnel here to ask questions when there is something that they do not understand	17.3%	15.0%	67.7%	3.53	0.74
6. The physicians and nurses here work together as a well-coordinated team	0.0%	100.0%	0.0%	1.75	0.43
**Safety Climate**					
7. I would feel safe being treated here as a patient	0.0%	10.2%	89.8%	4.59	1.21
8. Medical errors are handled appropriately in this clinical area	0.0%	5.7%	94.3%	4.77	0.93
9. I know the proper channels to direct questions regarding patient safety in this clinical area	13.3%	5.2%	81.5%	4.57	0.93
10. I receive appropriate feedback about my performance	17.7%	0.0%	82.3%	4.65	0.76
11. In this clinical area, it is difficult to discuss errors	0.0%	92.2%	7.8%	1.31	1.07
12. I am encouraged by my colleagues to report any patient safety concerns I may have	11.0%	5.1%	83.9%	4.62	0.89
13. The culture in this clinical area makes it easy to learn from the errors of others	0.0%	100.0%	0.0%	1.77	0.42
**Job Satisfaction**					
14. My suggestions about safety would be acted upon if I expressed them to management	0.0%	100.0%	0.0%	1.46	0.50
15. I like my job.	13.2%	2.8%	84.0%	4.65	0.82
16. Working here is like being part of a large family	20.5%	0.0%	79.5%	4.59	0.81
17. This is a good place to work	17.2%	0.0%	79.5%	4.66	0.76
18. I am proud to work in this clinical area	50.0%	0.0%	50.0%	4.00	1.00
**Stress Recognition**					
19. Morale in this clinical area is high	22.0%	0.0%	78.0%	4.56	0.83
20. When my workload becomes excessive, my performance is impaired	0.0%	0.0%	100.0%	5.00	
21. I am less effective at work when fatigued	0.0%	0.0%	100.0%	5.00	
22. I am more likely to make errors in tense or hostile situations	15.3%	0.0%	84.7%	4.69	0.72
**Perceptions of Management**					
23. Fatigue impairs my performance during emergency situations (e.g., emergency resuscitation, seizure)	0.0%	0.0%	100.0%	5.00	
24. Management supports my daily efforts at unit	46.0%	20.9%	33.1%	3.24	1.45
25. Management does not knowingly compromise pt safety at unit	33.6%	50.0%	16.5%	2.33	1.49
26. Management is doing a good job in the unit	33.5%	49.5%	17.1%	2.35	1.50
27. Problem personnel are dealt with constructively by our unit	28.8%	57.0%	14.1%	2.14	1.45
28. I get adequate, timely info about events that might affect my work from the unit	31.8%	0.0%	68.2%	4.36	0.93
29. Management supports my daily efforts at hospital	31.1%	50.2%	18.7%	2.37	1.54
30. Management does not knowingly compromise pt safety at the hospital	71.1%	14.6%	14.3%	2.99	1.08
31. Management is doing a good job at the hospital	39.6%	0.0%	60.4%	4.21	0.98
32. Problem personnel are dealt with constructively at our hospital	30.8%	52.4%	16.8%	2.29	1.50
33. I get adequate, timely info about events that might affect my work from the hospital	23.6%	9.8%	66.6%	4.14	1.33
**Working Conditions**					
34. The levels of staffing in this clinical area are sufficient to handle the number of patients	30.6%	47.6%	21.8%	2.48	1.58
35. This hospital does a good job of training new personnel	37.7%	43.5%	18.8%	2.50	1.50
36. All the necessary information for diagnostic and therapeutic decisions is routinely available to me	29.2%	0.0%	70.8%	4.42	0.91

**Table 3 ijerph-19-09712-t003:** Comparison of Safety Attitudes Questionnaire (SAQ-SF) results in reference to socio-demographic features (** *p* < 0.001, *** *p* < 0.0001).

Position	Teamwork Climate (TC)		Safety Climate (SC)		Job Satisfaction (JS)		Stress Recognition (SR)		Perceptions of Management (PM)		Working Conditions (WC)	
	M	SD	M	SD	M	SD	M	SD	M	SD	M	SD
**Gender**												
**Women**	51.64	4.27	80.74	7.46	86.70	9.41	97.77	4.79	50.65	10.95	48.72	16.50
**Men**	49.27	4.85	81.58	9.54	89.31	8.57	99.25	2.98	52.58	10.31	47.00	10.90
**Total**	51.12	4.51	80.92	7.97	87.28	9.29	98.09	4.50	51.07	10.84	48.34	15.45
** *p* **	M < F ***				F < M ***		F < M ***		F < M **			
**Professional group**												
**Nurse-A**	51.57	4.05	79.40	8.15	87.67	9.82	98.77	3.72	53.31	11.50	47.19	15.83
**Paramedic-B**	50.69	4.63	83.26	6.54	84.90	8.47	96.39	5.67	46.44	7.61	54.15	15.75
**Physician-C**	50.51	5.24	81.98	8.15	88.81	8.39	98.28	4.32	50.73	10.59	44.99	12.48
**Total**	51.12	4.51	80.92	7.97	87.28	9.29	98.09	4.50	51.07	10.84	48.34	15.45
** *p* **	A > B, C **		A < B, C ***		B < A, C ***		B < A, C ***		B < A, C ***, C < A **		B > A, C ***	
**Seniority**												
**less than 6 months A**	50.89	6.12	84.80	5.60	90.57	7.25	100.00	0.00	50.11	10.80	48.76	18.92
**6–11 month B**	51.08	4.70	83.86	7.03	90.24	9.15	99.41	2.66	49.33	11.62	47.64	13.34
**1 to 2 yrs C**	51.20	3.60	77.20	9.65	85.63	8.59	97.00	5.35	51.90	10.55	46.81	16.92
**3 to 4 yrs D**	51.26	5.32	82.12	5.88	85.62	9.21	97.47	5.03	51.78	11.43	51.12	12.11
**5–10 yrs E**	50.86	3.55	79.30	8.53	88.32	8.61	98.03	4.57	52.63	9.96	45.20	14.57
**11 or more F**	51.33	2.66	79.69	7.55	85.31	11.62	97.85	4.73	48.71	9.99	49.90	17.15
**Total**	51.12	4.51	80.92	7.97	87.28	9.29	98.09	4.50	51.07	10.84	48.34	15.45
** *p* **			A > C, D, E, F B > C, D, E C < D, E, F D > E, F		A > C, D, F, A > E B > C, D, F C < E, D > E E < F		A > C, D, E, F B > C, D, E C < E D > E, F		A > E B < C, D, E C < F D < F E < F		A > E B < D C < D D < E E > F	
**Country**												
**France-A**	49.09	4.17	80.22	7.59	86.83	9.14	97.83	4.75	50.98	11.57	48.10	12.52
**Greece-B**	51.36	3.19	80.73	5.59	87.50	6.93	100.00	0.00	47.33	9.77	53.99	16.78
**Poland-C**	55.17	5.85	81.60	8.53	88.40	8.99	98.40	4.19	51.88	11.35	48.90	17.25
**Spain-D**	50.11	4.18	81.36	7.73	87.21	9.85	97.82	4.75	53.42	9.01	47.29	14.22
**Turkey-E**	52.13	3.29	80.74	9.13	87.11	9.77	97.67	4.87	49.42	11.80	47.09	17.26
**Total**	51.12	4.51	80.92	7.97	87.28	9.29	98.09	4.50	51.07	10.84	48.34	15.45
** *p* **	A < B, C, D, E ***, B < C, B > D C > D < E D < E						B > A, D, E B > C		B < A, C, D A < D		B > A, D, E B > C	

**Table 4 ijerph-19-09712-t004:** Distribution of positive (≥75) and negative (<75) results in different professional groups.

SAQ		Nurse-A (N = 561)		Paramedic-B (N = 253)		Physician-C (N = 247)		
	Category	*n*	%	*n*	%	*n*	%	*p*
**Teamwork Climate**	<75	561	100%	253	100%	247	98%	
	≥75							
**Safety Climate**	<75	120	21.39%	15	5.93%	32	12.65%	0.000001
	≥75	441	78.61%	238	94.07%	215	84.98%	
**Job Satisfaction**	<75	60	10.70%	24	9.49%	10	3.95%	0.004
	≥75	501	89.30%	229	90.51%	237	93.68%	
**Stress Recognition**	<75							
	≥75	561	100%	253	100.00%	247	97.63%	
**Perceptions of Management**	<75	549	0.000001	253	100.00%	247	97.63%	0.0005
	≥75	12	2.14%	0	0.00%	0	0.00%	
**Working Conditions**	<75	527	93.94%	212	83.79%	241	95.26%	
	≥75	34	6.06%	41	16.21%	6	2.37%	

**Table 5 ijerph-19-09712-t005:** Distribution of positive (≥75) and negative (<75) results in different country.

SAQ		France-A (N = 230)		Greece-B (N = 116)		Poland-C (N = 125)		Spain-D (N = 332)		Turkey-E (N = 258)		
	Category	*n*	%	*n*	%	*n*	%	*n*	%	*n*	%	*p*
**Teamwork** **Climate**	<75	230	100%	116	100%	125	100%	332	100%	258	100%	
	≥75											
**Safety Climate**	<75	40	17.39%	8	6.90%	24	19.20%	47	14.16%	48	18.60%	0.02
	≥75	190	82.61%	108	93.10%	101	80.80%	285	85.84%	210	81.40%	
**Job** **Satisfaction**	<75	15	6.52%	5	4.31%	5	4.00%	43	12.95%	26	10.08%	0.003
	≥75	215	93.48%	111	95.69%	120	96.00%	289	87.05%	232	89.92%	
**Stress** **Recognition**	<75											
	≥75	230	100.00%	116	100.00%	125	100.00%	332	100.00%	258	100.00%	
**Perceptions of** **Management**	<75	230	100.00%	116	100.00%	119	95.20%	332	100.00%	252	97.67%	0.00003
	≥75	0	0.00%	0	0.00%	6	4.80%	0	0.00%	6	2.33%	
**Working** **Conditions**	<75	230	100.00%	85	73.28%	113	90.40%	318	95.78%	234	90.70%	0.000001
	≥75	0	0.00%	31	26.72%	12	9.60%	14	4.22%	24	9.30%	

**Table 6 ijerph-19-09712-t006:** Pearson’s r correlations between SAQ categories.

SAQ	M	SD	K	A	Teamwork Climate	Safety Climate	Job Satisfaction	Stress Recognition	Perceptions of Management	Working Conditions
**Teamwork** **Climate**	51.12	4.51	0.0	1.3						
**Safety Climate**	80.92	7.97	−0.8	0.0	0.03					
**Job Satisfaction**	87.28	9.29	−0.4	−0.3	0.02	0.04				
**Stress Recognition**	98.09	4.50	−1.4	1.4	−0.01	0.10	0.27			
**Perceptions of** **Management**	51.07	10.84	−0.1	−0.8	0.01	−0.06	−0.02	−0.01		
**Working** **Conditions**	48.34	15.45	0.0	−0.6	0.11	−0.01	0.25	0.02	−0.06	

**Table 7 ijerph-19-09712-t007:** Multiple regression results.

	Multiple R-Spearman	Multiple R Square [%]	R Square Change	*p*
**Teamwork Climate**	**R^2^ = 0.19, *p* < 0.000**			
Poland	0.328	11%	11%	0.000
Turkey	0.385	15%	4%	0.000
France	0.401	16%	1%	0.000
5–10 yrs	0.415	17%	1%	0.000
11 or more	0.426	18%	1%	0.001
Spain	0.437	19%	1%	0.000
**Safety Climate**	**R^2^ = 0.15, *p* < 0.000**			
1 to 2 yrs	0.231	5%	5%	0.000
Nurse	0.301	9%	4%	0.000
less than 6 months	0.344	12%	3%	0.000
6–11 month	0.367	13%	2%	0.000
3 to 4 yrs	0.388	15%	2%	0.000
Physician	0.393	15%	0%	0.025
**Job Satisfaction**	**R^2^ = 0.07, *p* < 0.000**			
Paramedic	0.143	2%	2%	0.000
less than 6 months	0.204	4%	2%	0.000
6–11 month	0.237	6%	1%	0.000
5–10 yrs	0.256	7%	1%	0.001
**Stress Recognition**	**R^2^ = 0.11, *p* < 0.000**			
Paramedic	0.211	4%	4%	0.000
less than 6 months	0.274	8%	3%	0.000
Greece	0.305	9%	2%	0.000
6–11 month	0.322	10%	1%	0.000
France	0.330	11%	1%	0.013
**Perceptions of Management**	**R^2^ = 0.10, *p* < 0.000**			
Paramedic	0.239	6%	6%	0.000
Spain	0.277	8%	2%	0.000
Greece	0.297	9%	1%	0.000
Physician	0.314	10%	1%	0.000
**Working Conditions**	**R^2^ = 0.07, *p* < 0.000**			
Paramedic	0.211	4%	4%	0.000
Greece	0.254	6%	2%	0.000
3 to 4 yrs	0.272	7%	1%	0.001

## Data Availability

A dataset will be made available upon request to the corresponding authors one year after the publication of this study. The request must include a statistical analysis plan.

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
