# Peer review of "The Perception of the Patient Safety Climate by Health Professionals during the COVID-19 Pandemic—International Research"

_ijerph, 2022, doi:10.3390/ijerph19159712_

Round 1

Reviewer 1 Report

Interesting study on the lived reality - congrats.

The study states that the instrument was validated in several care contexts. But was it valid in the different countries where it was used? (major concern)

If the exclusion criteria are the opposite of the inclusion criteria, it makes no sense to put.

Although the results say which countries the questionnaire was sent to, it makes sense to mention which countries were in the method. In the abstract, it should be clear whether there are differences between the countries analyzed - something that is not mentioned.

The last column of Table 2 is not perceptible.

The conclusion by key points seems to be different from usual but enlightening.

Author Response

Dear                                                                                                                                                                                             30. July 2022
Editor and Reviewers
International Journal of Environmental Research and Public Health

RESPONSES TO THE COMMENTS

Title: “The Perception of the Patient Safety Climate by Health Professionals During COVID-19 Pandemic - International Research”

We want to express our great appreciation to You and the reviewers for taking the time and effort necessary to review our manuscript entitled:

The Perception of the Patient Safety Climate by Health Professionals During COVID-19 Pandemic - International Research.

We carefully considered your comments. They helped us a lot; we appreciate Your patience and willingness to help us to make this manuscript better. Herein, we explain how we revised the paper for a second time based on Your comments and recommendations. All changes are listed in the file below. We have accepted all your suggestions.

Yours sincerely

Sabina Krupa

Reviewer 1

COMMENTS FROM THE EDITOR

CHANGES MADE

The study states that the instrument was validated in several care contexts. But was it valid in the different countries where it was used? (major concern)

We used the English version of the Safety Attitudes Questionnaire (SAQ-SF) in the study. Knowledge of the English language was one of the conditions for participation in the study. The Safety Attitudes Questionnaire demonstrated good psychometric properties The ρ value for the SAQ in this sample was 0,90, indicating strong reliability of the SAQ.

The SAQ was cross-culturally validated in different languages including Turkish, Polish, Spanish, Greek, and also Albanian, Arabic, Danish, Chinese, Croatian, Dutch, German, Italian, Norwegian, Portuguese, Slovenian, and Swedish. All these studies have shown that the SAQ possesses good psychometric properties in different languages. 

We added red in the text.

The Safety Attitudes Questionnaire demonstrated good psychometric properties. Composite scale reliability for the SAQ was assessed via Raykov's ρ coefficient. The ρ value for the SAQ in this sample was 0.90, indicating strong reliability of the SAQ [11].

If the exclusion criteria are the opposite of the inclusion criteria, it makes no sense to put.

Thank you for the suggestion, we have made changes to the document.

Although the results say which countries the questionnaire was sent to, it makes sense to mention which countries were in the method. In the abstract, it should be clear whether there are differences between the countries analyzed - something that is not mentioned.

We got a few questionnaires from Italy (24), Germany (57), Switzerland (38), Austria (17), Croatia (67) and Sweden (12) but they were not included in the analysis due to the small number. For the analysis, we chose those countries in which we obtained at least 100 correctly completed questionnaires.

We added red in the text.

In Poland, Spain, France, Turkey, and Greece healthcare workers scored highest in stress recognition (SR). In Poland, Spain, France, and Turkey they assessed working conditions (WC) the worst, while in Greece the perception of management (PM) had the lowest result.

The last column of Table 2 is not perceptible.

Thank you for the suggestion, the table has been revised to be more readable.

The conclusion by key points seems to be different from usual but enlightening.

Thank you for your comment.

Reviewer 2

COMMENTS FROM THE EDITOR

CHANGES MADE

1. Since this is a cross-sectional study, it will be helpful if a structured reporting guideline such as the STROBE is used in reporting this study. 

We added red in the text and Supplementary material 1

The study was conducted according to the Strengthening the Reporting of Observational Studies in Epidemiology (STROBE) criteria (Supplementary material 1)

2. Although the authors present detailed information regarding the SAQ tool, its psychometric properties are not reported. For example, what is the Cronbach's alpha?

We added red in the text.

The Safety Attitudes Questionnaire demonstrated good psychometric properties. Composite scale reliability for the SAQ was assessed via Raykov's ρ coefficient. The ρ value for the SAQ in this sample was 0.90, indicating strong reliability of the SAQ [11].

3. For the exclusion criteria, how about staff who were on leave during the period of the study?

Only professionally active medical personnel took part in the study.

4. Regarding section 2.6, the term "Pearsona" should read as "Pearson" (page 4). 

Thank you, the typing error has been corrected.

5. Please correct this figure "1.454" as "1,454" (page 5); the same applies to the abstract. 

Thank you, the typing error has been corrected.

6. Please check the heading of table 1: the figure 1161 should be corrected as 1061.

Thank you, the typing error has been corrected.

Reviewer 2 Report

The authors present a cross-sectional study regarding healthcare staff perceptions of patient safety climate during the COVID-19 pandemic. Overall, the study is interesting, and the multi-country nature of the study is particularly noteworthy. Please see below few comments for your consideration: 

1. Since this is a cross-sectional study, it will be helpful if a structured reporting guideline such as the STROBE is used in reporting this study. 

2. Although the authors present detailed information regarding the SAQ tool, its psychometric properties are not reported. For example, what is the Cronbach's alpha?

3. For the exclusion criteria, how about staff who were on leave during the period of the study?

4. Regarding section 2.6, the term "Pearsona" should read as "Pearson" (page 4). 

5. Please correct this figure "1.454" as "1,454" (page 5); the same applies to the abstract. 

6. Please check the heading of table 1: the figure 1161 should be corrected as 1061. 

Author Response

Dear                                                                                                                                                                                             30. July 2022
Editor and Reviewers
International Journal of Environmental Research and Public Health

RESPONSES TO THE COMMENTS

Title: “The Perception of the Patient Safety Climate by Health Professionals During COVID-19 Pandemic - International Research”

We want to express our great appreciation to You and the reviewers for taking the time and effort necessary to review our manuscript entitled:

The Perception of the Patient Safety Climate by Health Professionals During COVID-19 Pandemic - International Research.

We carefully considered your comments. They helped us a lot; we appreciate Your patience and willingness to help us to make this manuscript better. Herein, we explain how we revised the paper for a second time based on Your comments and recommendations. All changes are listed in the file below. We have accepted all your suggestions.

Yours sincerely

Sabina Krupa

Reviewer 2

COMMENTS FROM THE EDITOR

CHANGES MADE

1. Since this is a cross-sectional study, it will be helpful if a structured reporting guideline such as the STROBE is used in reporting this study. 

We added red in the text and Supplementary material 1

The study was conducted according to the Strengthening the Reporting of Observational Studies in Epidemiology (STROBE) criteria (Supplementary material 1)

2. Although the authors present detailed information regarding the SAQ tool, its psychometric properties are not reported. For example, what is the Cronbach's alpha?

We added red in the text.

The Safety Attitudes Questionnaire demonstrated good psychometric properties. Composite scale reliability for the SAQ was assessed via Raykov's ρ coefficient. The ρ value for the SAQ in this sample was 0.90, indicating strong reliability of the SAQ [11].

3. For the exclusion criteria, how about staff who were on leave during the period of the study?

Only professionally active medical personnel took part in the study.

4. Regarding section 2.6, the term "Pearsona" should read as "Pearson" (page 4). 

Thank you, the typing error has been corrected.

5. Please correct this figure "1.454" as "1,454" (page 5); the same applies to the abstract. 

Thank you, the typing error has been corrected.

6. Please check the heading of table 1: the figure 1161 should be corrected as 1061.

Thank you, the typing error has been corrected.
